# Physical-Layer Security in Power-Domain NOMA Based on Different Chaotic Maps

**DOI:** 10.3390/e25010140

**Published:** 2023-01-10

**Authors:** Mariam Abu Al-Atta, Karim A. Said, Mohamed A. Mohamed, Walid Raslan

**Affiliations:** 1Electronics and Communications Department, Faculty of Engineering, Delta University for Science and Technology, Gamsaa 35712, Egypt; 2Electronics and Communication Engineering Department, Faculty of Engineering, Mansoura University, Mansoura 35516, Egypt

**Keywords:** nonorthogonal multiple access, physical-layer security, power domain, downlink, encryption, chaos, logistic map, Arnold’s cat map, decryption, hyperchaotic DNA, Hénon map

## Abstract

Nonorthogonal multiple access (NOMA) is a relevant technology for realizing the primary goals of next-generation wireless networks, such as high connectivity and stability. Because a rising number of users are becoming connected, user data security has become a critical issue. Many chaotic communication systems have been established to address this important issue via exhibition of affordable physical-layer-security solutions. In this study, we propose a chaotic downlink NOMA (C-DL-NOMA) system over the additive white Gaussian noise and Rayleigh-fading channels to enhance the security of the DL-NOMA system. The proposed algorithm is based on a coherent analog modulation technique that combines various chaotic maps for chaotic masking of encrypted data. On the transmitter, chaotic encryption was used for transmitted data with fixed power-allocation-level control, whereas on the receiver, successive interference-cancellation demodulation was utilized to detect multiple users, after which chaotic decryption was performed. Simulation results were evaluated based on security analyses, such as statistical analysis (histogram and correlation analyses and information entropy), bit-error-rate performance, and achievable-data-rate performance. According to these security analyses and numerical results, the proposed C-DL-NOMA system outperformed traditional unencrypted NOMA systems.

## 1. Introduction

Nonorthogonal multiple access (NOMA) is a multiple-access technology for next-generation (5G) mobile networks. It is a nonorthogonal multiplexing method that allows users to be multiplexed in the power domain, which had previously been neglected by wireless mobile systems. Many users’ signals are combined at the transmitter and then segregated via successive interference cancellation (SIC) at the receiver [1]. In recent years, the multiple access (MA) system has been considered a significant high-tech advancement that has defined each generation of wireless communication networks. From the 1G mobile communication system to the 4G system, orthogonal MA (OMA) has been extensively used in current wireless systems. OMA systems include the widely used frequency-division MA for 1G, time-division MA for 2G, code-division MA (CDMA) for 3G, and orthogonal-frequency-division MA for 4G. Notably, only a limited number of users are multiplexed orthogonally within the frequency, time, or code domain in these OMA methods [2]. Several promising technologies have been proposed in recent years to address this critical issue. One such technology is nonorthogonal multiple access (NOMA): a way of serving numerous users from a single wireless resource. NOMA can be accomplished in different domains, including power, code, and others. For participation in the total resource, code-domain NOMA uses user-specific spreading sequences, whereas power-domain NOMA uses channel-gain changes among users for multiplexing through power allocation [3]. Owing to the open propagation environment, wireless networks are prone to interception from unauthorized receivers, and the enormous rise in NOMA users and related data traffic could result in more information security breaches, thereby making standard wireless security techniques depend on upper-layer cryptography. Because use of upper-layer security in NOMA systems with power-limited devices is ineffective due to cost and complexity restrictions, a reliable and economical physical-layer security (PLS) strategy that depends on channel-coding methods has been developed [4,5,6]. This strategy improves the performance gap between the intended user and the eavesdropper, using basic channel features such as noise, fading, diversity, and interference [4]. In addition, when combined with encryption-based systems, it can increase the entire system’s security [7,8]. Several types of encryption algorithm exist, each designed for a specific purpose. When existing algorithms become insecure, new ones are developed. The Data Encryption Standard, the Advanced Encryption Standard, Blowfish encryption, and Rivest Cipher 4 are some of the most well-known cryptographic algorithms. Some recent encryption methods have been shown to be untrustworthy for ciphering [9]. Owing to its sensitivity to control parameters and initial conditions, chaos-based encryption is inherent in chaotic methods of meeting security demands.

Several studies have described various chaos-based secure communication (CBSC) methods that offer cost effective and robust PLS for multiuser wireless applications. Chaos-based, secure power-domain NOMA was introduced in [6] for the uplink large-scale and Rician channels, with effective dynamic power control for transmitted multilevel chaos-shift-keying signals. For multiuser detection, an advanced receiver scheme that depends on SIC and chaos demodulation is suggested. The authors in [7] introduced a downlink (DL) chaotic NOMA (C-NOMA) method that used the C-MIMO idea to achieve high-capacity allocation and PLS. In [8], an uplink C-NOMA transmission technique, in which C-MIMO was used with NOMA to improve system throughput while providing PLS to other cell users and an eavesdropper, was proposed. NOMA successfully replicated the impacts of channel coding and PLS in C-MIMO. The authors in [10] offered a DL-C-NOMA transmission technique that achieves better capacity while providing PLS that is resistant to eavesdroppers and other users without initial keys. They used the C-MIMO scheme principle in NOMA to achieve this aim. Furthermore, while a low-complexity decoding technique for C-NOMA, which combined partial maximum-likelihood sequence estimation and the popular NOMA decoding method of SIC, was proposed in [11], an effective chaos-based NOMA (CB-NOMA) system for secure wireless communication was proposed in [12] over a Rayleigh-fading environment. Equal power allocation for linked users with a chaotic code domain is used in an integrated CB-NOMA design. Various chaotic-code-formation schemes with varying levels of security, implementation, and complexity have been studied for signaling via chaos shift keying (CSK). 

The aforementioned studies indicated that different coherent modulation techniques, such as chaos modulation [6,11], CSK [6], and chaos-based code-domain MA [12], have been used over NOMA. To improve the security of the DL-NOMA scheme, a chaotic downlink NOMA (C-DL-NOMA) system is proposed in this study. The proposed system uses a coherent analog modulation technique that combines various chaotic maps for chaotic masking (CM) of encrypted data and SIC decoding for complexity reduction over additive white Gaussian noise (AWGN) and Rayleigh-fading single-input–single-output (SISO) DL power-domain NOMA channels. This study’s main contributions are summarized below.

The effects of various hybrid chaotic maps on DL-NOMA performance are investigated using security analysis represented by statistical analyses, which include histogram and correlation analyses.The proposed C-DL-NOMA system provides robust PLS and a low bit error ratio (BER) with fixed power-allocation-level control, depending on the distance from the base station (BS).The proposed C-DL-NOMA system is compared with traditional unencrypted NOMA in terms of BER and achievable data rate.

The rest of this paper is organized as follows: Section 2 explains the DL-NOMA system model; Section 3 presents a C-DL-NOMA system model and different chaotic maps; Section 4 explains and discusses the simulation results; and finally, Section 5 concludes this study.

## 2. System Model 

This section describes the SISO DL power-domain NOMA system model (Figure 1). One-cell NOMA combines M users and one BS, each with one antenna. The channel response is perfectly known at the base station. The BS transmits the superposed signal to all mobile users on the transmitter side of the DL-NOMA system; this is a superposition of several users’ required signals with varying fixed power-coefficient allocations based on their distances from the BS [13]. The SIC procedure is supposed to be implemented sequentially on each user’s receiver side until the user’s signal is retrieved. The user with the greatest transmission power retrieves its signal with no execution of any SIC procedure and treats the other users’ signals as noise. However, other users must perform the SIC procedure. In the SIC procedure, every user’s receiver first discovers signals that are stronger than the user’s required signal. These signals are then subtracted from the received signal, and the procedure is repeated until the user’s required signal is identified. Finally, through consideration of users with lower power coefficients as noise, every user decodes its signal [5].

As in [13], the transmitted superposed signal at the BS is expressed as follows:(1)$$x_{s}=\overset{M}{\underset{i=1}{\sum }}\sqrt{a_{i}P_{t}\;}x_{i}$$
where $x_{i}$ is the data of user i ($U_{i}$) with unit energy, $P_{t}$ is the transmitted power at the BS, and $a_{i}$ is the power-allocation coefficient for user i; the allocation of power levels depends on the distance of every user so that $a_{i}^{\prime }=\frac{d_{i}}{d_{max}}$ (i=1, 2, …M), where $a_{i}^{\prime }$ is the absolute power factor and $d_{i}$ is the distance between the BS and the ith user, with $d_{max}=200\;m$ (a typical BS radius in 5G networks). The power factors are normalized between 0 and 1 as follows: $a_{i}=\frac{a_{i}^{\prime }}{\sum_{v=1}^{M}a_{v}^{\prime }}$ (v=1, 2, …M) [14]. In other words, $\overset{M}{\underset{i=1}{\sum }}a_{i}=1$ and $1>a_{1}\geq \;a_{2}\;\geq \;\ldots \;a_{M}>0$ because without loss of generality, the channel gains are supposed to be arranged as ${\left|h_{1}\right|}^{2}\;\leq \;{\left|h_{2}\right|}^{2}\;\leq \;\ldots \;{\left|h_{M}\right|}^{2}$, where $h_{i}$ is the Rayleigh-fading channel coefficient among the BS and the $m^{\mathrm{th}}\;$ user.

As in [13], the received signal at the $m^{\mathrm{th}}$ user can be written as follows: (2)$$y_{m}=h_{m}x_{s}+n_{m}=h_{m}\overset{M}{\underset{i=1}{\sum }}\sqrt{a_{i}P_{t}}x_{i}+n_{m}$$
where $n_{m}\;$ is complex, zero-mean AWGN with a variance of $\sigma^{2}$ and is denoted as $n_{m}\;∼\;CN\;\left(0,\sigma^{2}\right).$

Notably, users use SIC to obtain desired signals from the received signal, $y_{m}$. In particular, $U_{m}$ starts by decoding the stronger $U_{1}$ signal through treatment of the remaining part of the signal as interference. Thereafter, to acquire a signal free of the $U_{1\;}\mathrm{signal}$, $U_{m}$ remodulates the decoded signal and removes it from the received signal, $y_{m}$. Executing an identical process, $U_{m}$ progressively cancels the whole signals corresponding to users $U_{2}$, $U_{3}$, …, $U_{m-1}$ from the received signal, $y_{m}$, lastly decoding its signal through treatment of the rest of the signal parts, corresponding to users $U_{m+1}$, $U_{m+2}$, …, $U_{M}$, as interference [15].

From Equation (2), the $m^{\mathrm{th}}$ user’s signal to interference and noise ratio (SINR) is used to discover the $j^{th}$ user, j ≤ m, with j ≠ M, expressed as follows [13]:(3)$$SINR_{j\to m}=\frac{a_{j}\rho_{s}{\left|h_{m}\right|}^{2}}{\rho_{s}{\left|h_{m}\right|}^{2}\sum_{i=j+1}^{M}a_{i}+1}$$
where $\rho_{s}=P_{t}/\sigma^{2}$ indicates the signal-to-noise ratio (SNR). To obtain the $m^{\mathrm{th}}$ user’s required data, the SIC procedure is performed on the signal of user j ≤ m. Thus, the $m^{\mathrm{th}}$ user’s SINR is expressed as follows [13]:(4)$$SINR_{m}=\frac{a_{m}\rho_{s}{\left|h_{m}\right|}^{2}}{\rho_{s}{\left|h_{m}\right|}^{2}\sum_{i=m+1}^{M}a_{i\;}+1}$$

Then, the $M^{\mathrm{th}}$ user’s SINR is given with [13]: (5)$$SINR_{M}=a_{M}\rho_{s}{\left|h_{M}\right|}^{2}$$

After the SINR terms of DL-NOMA are acquired, when all of the symbols ($x_{1}$, $x_{2}$, …, $x_{m-1}$) have been accurately decoded, the achievable rate at $U_{m}$ for decoding signal $x_{m}$ can be gained as follows [13]:(6)$$R_{m}^{DL}=\log_{2}(1+SINR_{m})=\log_{2}(1+\frac{a_{m}\rho_{s}{\left|h_{m}\right|}^{2}}{\rho_{s}{\left|h_{m}\right|}^{2}\sum_{i=m+1}^{M}a_{i}+1})$$

Thus, the sum rate of DL-NOMA is described as follows [13]:(7)$$\begin{array}{l} R_{sum}^{DL}=\overset{M}{\underset{m=1}{\sum }}\log_{2}(1+SINR_{m})\;\\ =\overset{M-1}{\underset{m=1}{\sum }}\log_{2}(1+\frac{a_{m}\rho_{s}{\left|h_{m}\right|}^{2}}{\rho_{s}{\left|h_{m}\right|}^{2}\sum_{i=m+1}^{M}a_{i}+1})+\log_{2}\left(1+a_{M}\rho_{s}{\left|h_{M}\right|}^{2}\right)\;\\ =\overset{M-1}{\underset{m=1}{\sum }}\log_{2}(1+\frac{a_{m}}{\sum_{i=m+1}^{M}a_{i}+1/\rho_{s}{\left|h_{m}\right|}^{2}})+\log_{2}\left(1+a_{M}\rho_{s}{\left|h_{M}\right|}^{2}\right)\; \end{array}$$

With a high SNR that is as $\rho_{s}\;\to \;\infty$, the sum rate of DL-NOMA becomes [13]
(8)$$R_{sum}^{DL}\approx \;\overset{M-1}{\underset{m=1}{\sum }}\log_{2}(1+\frac{a_{m}}{\sum_{i=m+1}^{M}a_{i}})+\log_{2}\left(\rho_{s}{\left|h_{M}\right|}^{2}\right)\;\approx \log_{2}\left(\rho_{s}{\left|h_{M}\right|}^{2}\right)$$

## 3. The Proposed C-DL-NOMA System Model

Having introduced the DL power-domain NOMA communication system, we want to support the security of data transmission through it via encryption of this data (Figure 2) using different chaotic maps. In the area of CBSC, chaotic signals are frequently employed for various modulation techniques, including chaos on–off keying (COOK), CSK, chaos-parameter modulation, and CM. In this study, we applied CM modulation to DL power-domain NOMA using hybrid chaotic maps. In this technique, plain data are added to the chaotic signal sequence, after which the produced data are transmitted through the NOMA channel. The plain data can be restored through subtraction of the regenerated chaotic signal sequence from the received data. We now explain the chaotic maps used in our proposed C-DL-NOMA system and how we employ them to encrypt the data. Chaotic maps have attractive characteristics, such as easy generation that utilizes low-cost circuits, a noise-like spectrum, unpredictable behavior, sensitivity to initial conditions, and rapid iteration. Therefore, chaos-based data encryption techniques are suitable for real-time applications. Some existing chaotic maps are discussed below.

### 3.1. Logistic Chaotic Map

The logistic map is very basic and is often used as a normal chaotic map. It appears to be quite simple and predictable, but its dynamic behavior is extremely complicated. The logistic map is defined in [16] as:(9)$$\chi_{n+1}=r\chi_{n}\left(1-\chi_{n}\right)$$
where n=0, 1, 2, 3, … ($0<\chi_{0}<1$ and 0<r ≤ 4).

Here, $\chi_{0}$ and r determine the sequences generated with the logistic map (which is the initial value of χ). If even one of the two values changes slightly, the result will differ significantly. The system will exhibit varied properties depending on deviation level [17]. When 3.56994<r ≤ 4 is reached, chaotic behavior occurs, as indicated in Figure 3 [18].

**Encryption and Decryption Process**: Assume the following parameters: $\chi_{0}=0.1$ and r=4. When this is iterated, we obtain $\chi_{1}$, $\chi_{2}$, …,$\;\chi_{n}$ for the N value. This is a chaotic one-dimensional sequence (A).To encrypt the MN-size original image, a matrix of the same size must be created through transformation of the one-dimensional sequence (A) to a two-dimensional sequence (B), where B is the cipher.The original image is encrypted using an XOR cipher.The encrypted image is decrypted using the XOR cipher.


**Encryption and Decryption Algorithm:**
Input: original image, χ0, and r. Output: encrypted image.**Step 1.** Transform the original image of a size of M×N pixels with an array of P=pi,j, where (1<i<M) and (1<j<N). **Step 2.** Next, convert the pixel values to unsigned integers in the range of 0 to 255, using mod operation Pm=mod {P,255}.**Step 3.** Generate the n number of chaotic sequence A={χ1 , χ3 , χ3 …, χn} in the range of 0 to 1 using the logistic map in Equation (9), with an initial condition of χ0=0.1 and taking the parameter of r=4. **Step 4.** Transform the chaotic sequence, A, to an array, B, of a size of M×N. **Step 5.** Next, convert B into unsigned integers in the range of 0 to 255, using mod operation Bm=mod {B,255}.**Step 6.** Encrypted image = Pm ⊕ Bm.**Step 7.** The decryption process is identical to that of encryption in reverse order.

### 3.2. Hénon Chaotic Map

The Hénon chaotic map is a discrete-time dynamic system that depends on two parameters, a and b, which are essential because they control the system’s dynamic behavior. The classical Hénon map uses the values a=1.4  and b=0.3, which results in chaotic behavior. This map can be chaotic, be intermittent, or converge to a periodic orbit for alternative values of a and b [19]. The Hénon chaotic map is formed using the following equations:(10)$$x_{n+1}=1-\;ax_{n}^{2}+y_{n}$$
(11)$$y_{n+1}=\;bx_{n}$$
where (*n* = 0, 1, 2, …); $x_{n}$ and $y_{n}$ are the current point positions; and $x_{n+1}$ and $y_{n+1}$ are the next point positions. Initial points $x_{1}$ and $y_{1}$ [20] act as a symmetric key for a chaotic cryptographic system that is utilized for encryption and decryption at the sender’s and receiver’s terminals. Since the Hénon map is deterministic, decryption of the cipher image will reconstruct the original image on the receiver’s end with the same initial points: $x_{1}$ and $y_{1}$ [21].

Figure 4a [22] shows the  x − *y* plane graph of the Hénon map’s attractor. Figure 4b [23] shows the Hénon map’s bifurcation behavior, with b=0.3. The attractive set’s x coordinate is plotted along the horizontal axis for various values in the range of a from 0 to 1.4. The repetition sequence begins to divide into a two-period oscillation at a=0.35, which continues until a=0.85. Periodic-doubling bifurcation is another name for this dividing mechanism. In the approximate range of a from 0.85 to 1.1, periods are successively doubled. In the dark region, periodicity switches to chaotic behavior when a ≥ 1.1. The Hénon map is used to generate two random sequences that permute an image’s row and column positions, respectively, due to this chaotic tendency [24].

**Encryption and Decryption Process**:

Encryption Process, as shown in Figure 5:The number of pixels of the original image are extracted via multiplication of the image’s height and width.The Hénon chaotic map is used to scramble the original image’s pixels.The logistic chaotic map is used to generate key values or pseudorandom numbers.The encrypted image is obtained via the XOR operation, which is performed between the key values gained from the logistic map and the pixel values obtained from the scrambled image that resulted from the operation with the Hénon chaotic map.

Decryption Process, as shown in Figure 5:The encrypted image’s pixels are extracted via measurement of the image’s height and width.The XOR operation is performed between the pixel values obtained from the encrypted image and the key values gained from the logistic map.The decrypted image is obtained via use of the Hénon chaotic map to scramble the pixels of the image that resulted from the XOR operation.

**Figure 5 entropy-25-00140-f005:**
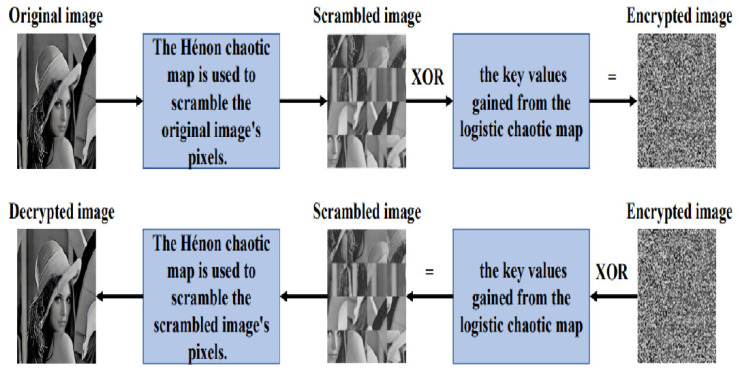
Encryption and decryption processes using the Hénon chaotic map.

### 3.3. Arnold’s Cat Chaotic Map 

Arnold’s cat map was designed as a confusion technique to randomize the pixel position of an image so that it does not appear the same. This can be achieved using Equation (12) [21]: (12)$$\left[\begin{array}{c} {ϗ}^{\prime }\\ {ϒ}^{\prime } \end{array}\;\right]=Ð\;\left[\begin{array}{c} ϗ\;\\ ϒ\; \end{array}\right]\left(mod\;ɳ\right)=\left[\begin{array}{cc} 1 & ĩ\\ ɉ & ĩɉ+1 \end{array}\right]\;\left[\begin{array}{c} ϗ\;\\ ϒ\; \end{array}\right]\left(mod\;ɳ\right)$$
where (${ϗ}^{\prime },{ϒ}^{\prime }$) is the new position of the original pixel position, (ϗ, ϒ); ĩ and ɉ are positive integers; and determinant (Ð)=1. Arnold’s cat map includes a unique hyperbolic fixed point. The linear transformation that describes the map is categorical: eigenvalues are irrational numbers, one higher and one lower than 1; hence, they correspond to expanding and contracting eigenspaces, which are also stable and unstable bifurcations. Because the matrix is identical, the eigenspace is orthogonal [20].

As shown in Figure 6, the process of mapping via returning to the unit square continuously shuffles the image (the phase space). The first step demonstrates shearing in the x and y directions, followed by image reassembly [25].


**Encryption Process:**
The number of pixels of the original image are extracted via multiplication of the image’s height and width.Arnold’s cat chaotic map is utilized to scramble the original image’s pixels.The logistic chaotic map is used to generate key values or pseudorandom numbers.The encrypted image is obtained via performance of the XOR operation using the key values obtained from the logistic map and the pixel values obtained from the scrambled image that resulted from the operation with Arnold’s cat chaotic map.


**Decryption Process**:The encrypted image’s pixels are extracted via measurement of the image’s height and width.The XOR operation is performed between the pixel values obtained from the encrypted image and the key values gained from the logistic map.The decrypted image is obtained via use of Arnold’s cat chaotic map to scramble the pixels of the image that resulted from the XOR operation.

### 3.4. Hyperchaotic Map 

A hyperchaotic system is a chaotic system with a positive Lyapunov exponent that exceeds 1, indicating that its chaotic dynamics are provided simultaneously in multiple directions [26]. Hyperchaos occurs in high-dimensional, nonlinear schemes with at least four dimensions. A high-dimensional chaotic system has a larger key space and more complex and unpredictable nonlinear behavior due to the increased number of state variables in a hyperchaotic system [27].

DNA (deoxyribonucleic acid) computing has recently been utilized in chaos-based image-encryption schemes because of its many advantages: enormous parallelism, massive storage, and ultralow power consumption [28].

As illustrated in Figure 7, the authors in [28] combined DNA sequencing and hyperchaotic sequencing for encryption, and we followed the same approach for our proposed C-DL-NOMA system.

The following nonlinear equations govern the hyperchaotic system [29], which they adopted [28]:(13)$${\dot{y}}_{1}=\alpha \left(y_{2}-y_{1}\right)+\lambda_{1}y_{4}$$
(14)$${\dot{y}}_{2}=\xi y_{1}-y_{1}y_{3}+\lambda_{2}y_{4}$$
(15)$${\dot{y}}_{3}=-\beta y_{3}+y_{1}y_{2}+\lambda_{3}y_{4}$$
(16)$${\dot{y}}_{4}=-\tau y_{1}$$
where the system’s control parameters are α, ξ, β, $\tau ,\;\lambda_{1},\;\lambda_{2},\;\mathrm{and}\;\lambda_{3}$. This system describes hyperchaotic performance when α= 35, ξ=35, β=3, τ=5, $\lambda_{1}=1$, $\lambda_{2}=0.2$, and $\lambda_{1}=0.3$.

The DNA sequence includes four bases of nucleic acid, which are permanently denoted by the letters G (Guanine), C (Cytosine), A (Adenine), and T (Thymine). “G” and “C” are complementary, as are “A” and “T”. We utilized two-bit binary digits to represent a DNA base, since the binary digits “0” and “1” are complementary. For the representation shown in [30], there are twenty-four different types of rule, and only eight of them satisfy the Watson–Crick complement rule [31]. DNA computing follows conventional binary addition and subtraction rules [32]. 

Four Steps to Generate the Hyperchaotic Sequence, i:To avoid negative effects and to increase security, the hyperchaotic system is pre-iterated $N_{0}$ times.After iteration of $N_{0}$ times, the process is repeated m×n times. We utilized ĵ to indicate the iteration index. For every iteration,  ĵ, four state values, $\left\{{\hat{y}}_{1}^{ĵ},{\hat{y}}_{2}^{ĵ},{\hat{y}}_{3}^{ĵ},{\hat{y}}_{4}^{ĵ}\right\},$ are saved.Every state value, ${\hat{y}}_{į}^{ĵ},$ is utilized to produce two various key values—${\left(v_{į}^{ɐ}\right)}^{ĵ}\in \left[0,255\right]$, (į=1,2,3,4) and $v_{į}^{ɐ}\in \left[0,255\right]$—through the iteration. They are determined as follows [28]:(17)$${\left(v_{į}^{ɐ}\right)}^{ĵ}=mod\;\left\{[\frac{\left[\left(\left|{\hat{y}}_{į}^{ĵ}\right|-\left\lfloor |{\hat{y}}_{į}^{ĵ}|\right\rfloor \right)*10^{15}\right]}{10^{8}}],255\;\right\},\;\;\;\;\;\;\;\;\;į=1,2,3,4\;$$
(18)$${\left(v_{į}^{ɓ}\right)}^{ĵ}=mod\left(\lfloor mod\;\left\{\frac{\left[\left(\left|{\hat{y}}_{į}^{ĵ}\right|-\left\lfloor |{\hat{y}}_{į}^{ĵ}|\right\rfloor \right)*10^{15}\right]}{10^{8}}\}\right\rfloor ,255\right),į=1,2,3,4$$

Here, mod(.) indicates the modulo operation and ⌊.⌋ indicates the flooring process, which rounds the element to the closest integer to negative infinity. These key values are connected with Equation (19) to become a vector, $v^{ĵ}$:(19)$$v^{ĵ}=v_{1}^{ɐ}+v_{2}^{ɐ}+v_{3}^{ɐ}+v_{4}^{ɐ}+v_{1}^{ɐ}+v_{2}^{ɐ}+v_{3}^{ɐ}+v_{4}^{ɐ}$$

4.After the entire iteration, these sequences are connected with the next equation to obtain k, $k=\left[v^{1},\;v^{2},\;\ldots ,v^{m\times n}\right].$ $k_{į}$ can be used to represent one element in k, į∈[1,8mn].

A global bit scrambling (GBS) system significantly enhances execution of image encryption. An input image, F, has eight bits and an intensity value between 0 and 255. The image’s intensity values are globally shuffled, bit by bit, to decrease the correlations among adjacent pixels. GBS also modifies the intensity value of each pixel, meaning it introduces pixel substitution simultaneously.

GBS is achieved using the following two steps:To obtain a one-dimensional binary sequence, ${ɓ}^{0}$, the intensity value per pixel is defined as binary digits, one by one. To obtain the index sequence, $k^{\hat{y}},$ in ascending order, the hyperchaotic sequence, k, is organized.${ɓ}^{0}$ is globally shuffled to be a one-dimensional binary sequence depending on the index sequence, $k^{\hat{y}}$; ${ɓ}_{į}^{1}\;=\;{ɓ}_{k_{į}^{\hat{y}}}^{0}$, į∈[1,8mn].

GBS is produced via a complex nonlinear relationship between the input and the encrypted images, which enhances security.

The Encryption and Decryption Process are carried out in seven steps:
m×n indicates the input image (F)’s size. The binary sequence, ${ɓ}^{1},$ is achieved through GBS operation on an image, F.According to the first DNA coding rule, ${ɓ}^{1}$ is encoded in a DNA sequence, $d^{1}$. The DNA addition to each element of $d^{1}$ is executed to obtain $d^{2}$ [28];
(20)$$d^{2}=\{{\;}_{d_{į}^{2}=d_{į-1}^{2}++d_{{į}^{\;}}^{1}}^{d_{1}^{2}=d_{0}++d_{1^{\;}}^{1}}\;\;\;\;\;\;\;\;į\;\in \left[2,4mn\right]$$
where ++ indicates the DNA addition process and $d_{0}$ is a specific initial value.A sequence, $k^{v}=\left[k_{1},k_{2},\ldots ,k_{mn}\right],\;$ is reduced from k, and the decimal sequence, $k^{v}$, is transformed into binary digits, $b^{k}$. According to the third DNA encoding rule, $b^{k}$ is encrypted into $d^{k}$. To obtain a sequence, $d^{3}$, DNA addition among $d^{2}$ and $d^{k}$ is performed.A threshold function, f(s), is expressed as follows [28]:(21)$$f\left(s\right)=\{\begin{array}{c} 0\;,0\leq \frac{s}{255}\leq 0.5\\ 1,\;0.5<\frac{s}{255}\leq 1 \end{array}$$

A cut sequence of k, $\left[k_{1},k_{2},\ldots ,k_{4mn}\right],$ is converted to a mask sequence, w, using Equation (21). The mask sequence, w, and $d^{3}$ are utilized to create $d^{4}$. To obtain $d_{į}^{4}$, the corresponding $d_{į}^{3}$ is complemented. If $w_{į}=1$, or else it is not varied. This is how the DNA sequence, $d^{4},$ is obtained.
5.To obtain the binary sequence, ${ɓ}^{2}$, $d^{4}$ is decoded using the first DNA coding rule.6.To obtain the encrypted binary sequence, ${ɓ}^{3}$, bitwise XOR is executed between ${ɓ}^{2}$ and ${ɓ}^{k}$.7.The encrypted binary sequence, ${ɓ}^{3}$, is converted to an encrypted image, T. The decryption process is identical to that of encryption in reverse order.

## 4. Simulation Results

This section discusses the performance of the proposed C-DL-NOMA system, based on different chaotic maps, through simulation results. This simulation was performed using MATLAB. The impacts of various chaotic maps assigned to Users 1, 2, and 3 on SISO DL power-domain NOMA performance, in terms of BER and some security analyses, were demonstrated via the simulation results. The proposed model for evaluating the quality of the system performance was simulated based on the parameters listed in Table 1.

### 4.1. Security Analysis

Regarding encryption, simply encrypting data is insufficient. Encrypted data should be extremely reliable. Some security analyses, such as statistical analyses represented by histogram and correlation analyses (Figure 8) and information entropy analysis (Table 2), must be performed to demonstrate high reliability.

#### 4.1.1. Statistical Analysis

Because the efficiency of an encryption system is determined from its ability to reject statistical attacks, statistical analyses are utilized to distinguish original data from encrypted data. Such statistical analyses are discussed below.

##### Histogram Analysis

A histogram distribution is a graphical representation of data distribution. It comprises the frequency of the data group. Several fields can use histogram analysis. Encryption is effective if the histogram’s distribution of the ciphered data has similar values. The closer the values of a data set are to each other, the more difficult it is to decode encrypted data. The histogram distribution in the image data is from left to right, that is, from dark to light colors. On the left side, dark colors prevailing indicate much distribution, whereas on the right side, light colors prevailing indicate much distribution. For this reason, the more uniform the distribution is, the more difficult it is to form a thought of the image or decode the ciphered data. Notably, hyperchaotic DNA has a better histogram distribution than do the logistic, Hénon, and Arnold’s cat maps because its ciphered data has similar values (Figure 8).

##### Correlation Analysis

Correlation analysis is commonly used in image-data security analysis. For correlation analysis to be useful, the relationship among variables must be linear. Testing the relationship among adjacent pixels can be used to execute correlation analysis.

The relationship among the variables on the plain image is linear, whereas the relation after analysis is nonlinear (scattered) with a highly complex distribution, and a correlation value close to zero indicates that the analysis result is good and there is no relation between the two images. An adjacent pair of pixels in any number is chosen at random from the image; the correlation coefficient of each pair is calculated using the following formulas [18]:(22)$$E\left(Ϫ\right)=\frac{1}{Ñ}\overset{N}{\underset{ɨ=1}{\sum }}{Ϫ}_{ɨ}$$
(23)$$E\left(Ϫ\right)=\frac{1}{Ñ}\overset{N}{\underset{ɨ=1}{\sum }}{Ϫ}_{ɨ}$$
(24)$$\mathrm{cov}\left(Ϫ,Ӌ\right)=\frac{1}{Ñ}\overset{N}{\underset{ɨ=1}{\sum }}\left({Ϫ}_{ɨ}-E\left(Ϫ\right)\right)\left({Ӌ}_{ɨ}-E\left(Ӌ\right)\right)$$
(25)$$r_{ϪӋ}=\frac{\mathrm{cov}\left(Ϫ,Ӌ\right)}{\sqrt{D\left(Ϫ\right)D\left(Ӌ\right)}}$$

Ϫ and Ӌ are the grayscale values of the two adjacent pixels in the image, and Ñ refers to the total number of pixels chosen from the image. In the horizontal direction, 3000 pairs of adjacent pixels are chosen at random from the images to display their adjacent pixel distribution maps. This indicates that input images have a strong correlation effect, whereas encrypted images have a weak correlation impact, confirming that robust correlation is absent among adjacent pixels in encrypted images. As shown in Figure 8, through the four chaotic encryption schemes, the hyperchaotic DNA algorithm has a good encryption effect that damages correlation and achieves the weakest correlation impact.

#### 4.1.2. Information Entropy Analysis

The information entropy of an image can reveal its information repeatability. The density of an eight-bit grayscale image has $2^{8}$ possible values, so its ideal information entropy value is 8. If the encoded image’s information entropy is closer to 8, it is closer to random distribution. Information entropy is described below [33]:(26)$$\mathrm{Information}\;\mathrm{entropy}\;\left(\underline{T}\right)=-\overset{2^{n_{b}}-1}{\underset{i=0}{\sum }}P\left({\underline{T}}_{\underline{i}}\right)\;\log_{2}P\left({\underline{T}}_{\underline{i}}\right)$$
where $\underline{T}$ indicates the encrypted image, $n_{b}$ indicates the total number of bits that represent the symbol ${\underline{T}}_{\underline{i}}$, and $P\left({\underline{T}}_{\underline{i}}\right)$ indicates the probability that ${\underline{T}}_{\underline{i}}$ will appear. The information entropy of the encrypted images is computed. As listed in Table 2, the encrypted images were close to a random source, and the proposed systems were sufficiently secured against entropy attacks.

### 4.2. Numerical Results

#### 4.2.1. Bit Error Rate (BER)

Figure 9 shows the BER versus the transmission SNR (dB) for each user in the four C-DL-NOMA schemes. Notably, the closest user (User 3) had the worst performance because two levels of SIC were performed for User 3, whereas User 2 outperformed User 3 because one level of SIC was performed at User 2. Otherwise, since User 1 demodulated the data directly without performing SIC, this user provided the best performance. In addition, we can see that the logistic scheme had the best BER performance and that the BERs of the Hénon and Arnold’s cat schemes were semisimilar, as they used the same method to generate key values. Finally, the hyperchaotic DNA scheme had the worst BER performance among the rest of the schemes because of the number of layers in the encryption and decryption processes. Compared with the 3 UE’s SUI-6 model in [14], we obtained better BER performance.

In Table 3, we chose the BER to be $10^{-3}$ to compare the achieved SNR (dB) per user in the four C-DL-NOMA schemes and the 3 UE’s SUI-6 model in [14]. The closest user (User 3), with the lowest power factor and two levels of SIC, achieved a BER of $10^{-3}$ at SNRs of 30 dB for NOMA with no encryption, 30.1 dB for the logistic scheme, 33.3 dB for the Hénon and Arnold schemes, and 33.6 dB for the hyperchaotic DNA scheme; the second-nearest user (User 2), with a single level of SIC, achieved a BER of $10^{-3}$ at SNRs of 26.8 dB for NOMA with no encryption, 26.9 dB for the logistic scheme, 29.5 dB for the Hénon and Arnold schemes, and 29.6 dB for the hyperchaotic DNA scheme; and the farthest user (User 1) achieved a BER of $10^{-3}$ at SNRs of 19.6 dB for NOMA with no encryption, 19.8 dB for the logistic scheme, 22 dB for the Hénon and Arnold schemes, and 22.18 dB for the hyperchaotic DNA scheme.

#### 4.2.2. Achievable and Sum Data Rates

Considering Equations (7) and (8), neither achievable nor sum data rates depend on transmitted data encrypted with different chaotic maps. Therefore, the rates for the four C-DL-NOMA schemes are similar. Figure 10 shows the simulation results of the achievable data rate per user and the sum rate versus the transmit SNR in dB. Notably, the achievable data rate of User 3 outperformed those of Users 1 and 2. Furthermore, the achievable data rates of Users 1 and 2 increased slightly in the low-SNR area and were saturated in the high-SNR area, but that of User 3 increased exponentially with the SNR because User 1 did not utilize SIC but only discovered the signal itself. Concurrently, User 2 needed to utilize first-order SIC first, after which User 3 should have utilized second-order SIC, implying that the effect of interference on User 1 was higher than on Users 2 and 3. Compared with the achievable rate for the three users in [34], we obtained a better data-rate performance.

## 5. Conclusions

In this study, we proposed a C-DL-NOMA method to support data transmission security through SISO DL power-domain NOMA over the AWGN and Rayleigh-fading channels. The proposed method is based on a coherent analog modulation technique for CM of encrypted data, where different hybrid chaotic maps, such as the logistic, Hénon, hyperchaotic, and Arnold’s cat maps, were considered. BER findings obtained utilizing various chaotic schemes confirmed the effectiveness of the proposed C-DL-NOMA model when compared to typical NOMA systems. Moreover, our DL power-domain NOMA with fixed allocated power factors based on the users’ distances from the BS achieved better achievable-data-rate performance. In addition, the security analysis revealed that our proposed C-DL-NOMA model has robust PLS to support data-transmission security. Future studies can incorporate any type of noncoherent modulation to obtain high connectivity with various quality services. The proposed encryption method can be applied to code-domain NOMA. It can also use dynamic power-allocation factors based on channel-state information values in C-DL-NOMA to offer user fairness and maximize the sum rate. 

## Figures and Tables

**Figure 1 entropy-25-00140-f001:**
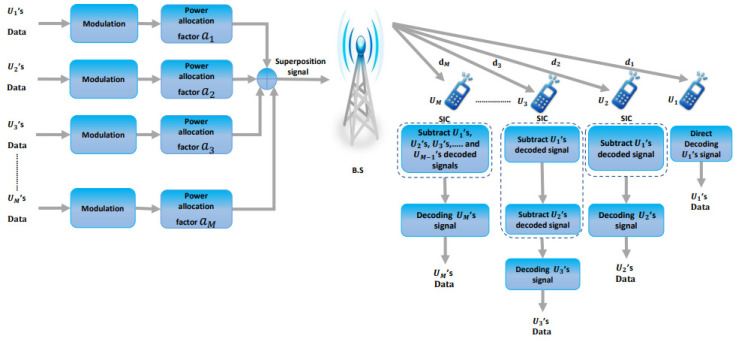
SISO DL-NOMA system model.

**Figure 2 entropy-25-00140-f002:**
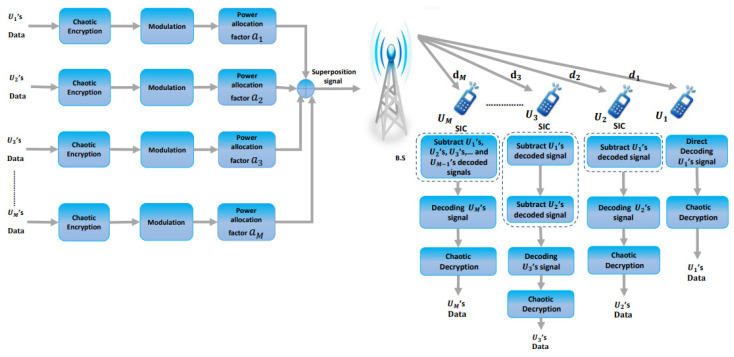
Proposed C-DL-NOMA system model.

**Figure 3 entropy-25-00140-f003:**
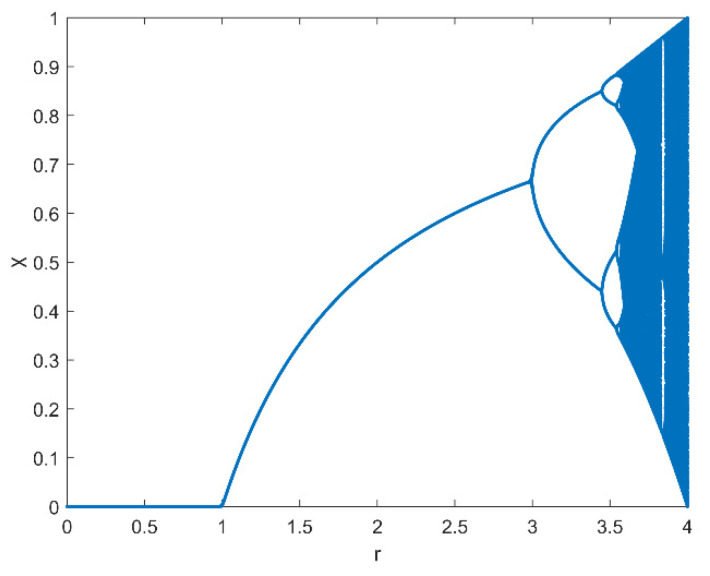
The logistic map bifurcation diagram.

**Figure 4 entropy-25-00140-f004:**
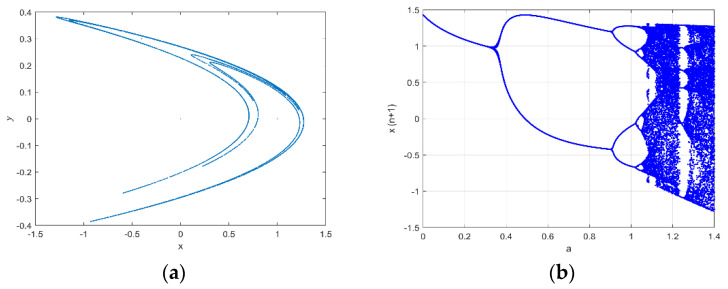
(**a**) The strange attractor of the Hénon map for *a* = 1.4 and *b* = 0.3. (**b**) Hénon map bifurcation diagram for *b* = 0.3.

**Figure 6 entropy-25-00140-f006:**
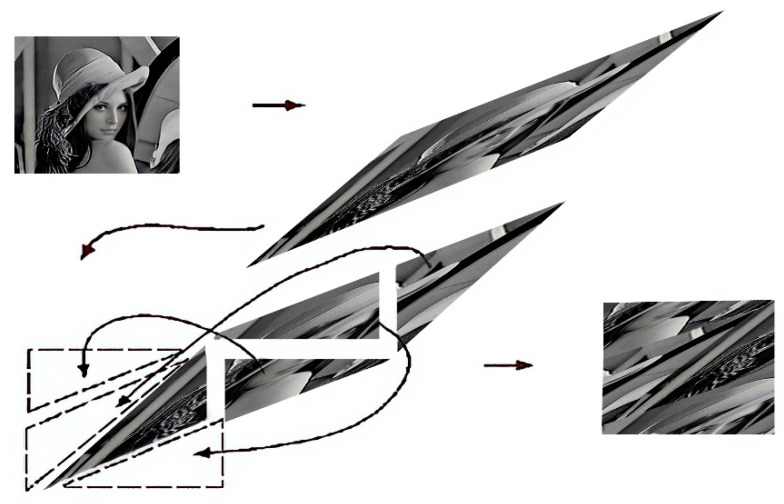
Graphical illustration of Arnold’s cat map.

**Figure 7 entropy-25-00140-f007:**
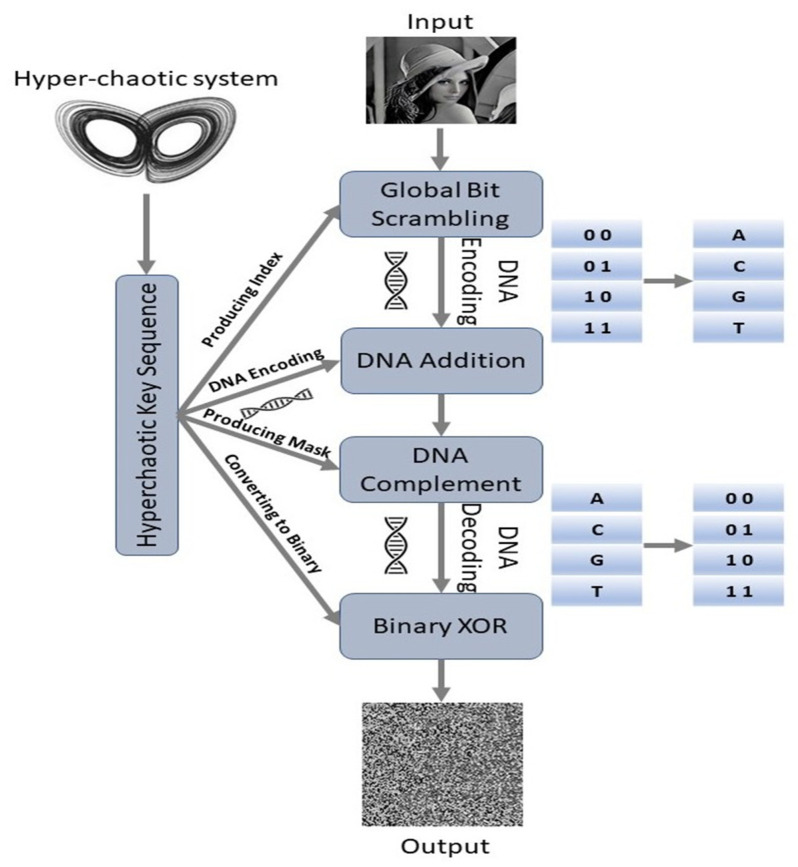
Schematic diagram of encryption using DNA sequencing and hyperchaotic sequencing.

**Figure 8 entropy-25-00140-f008:**
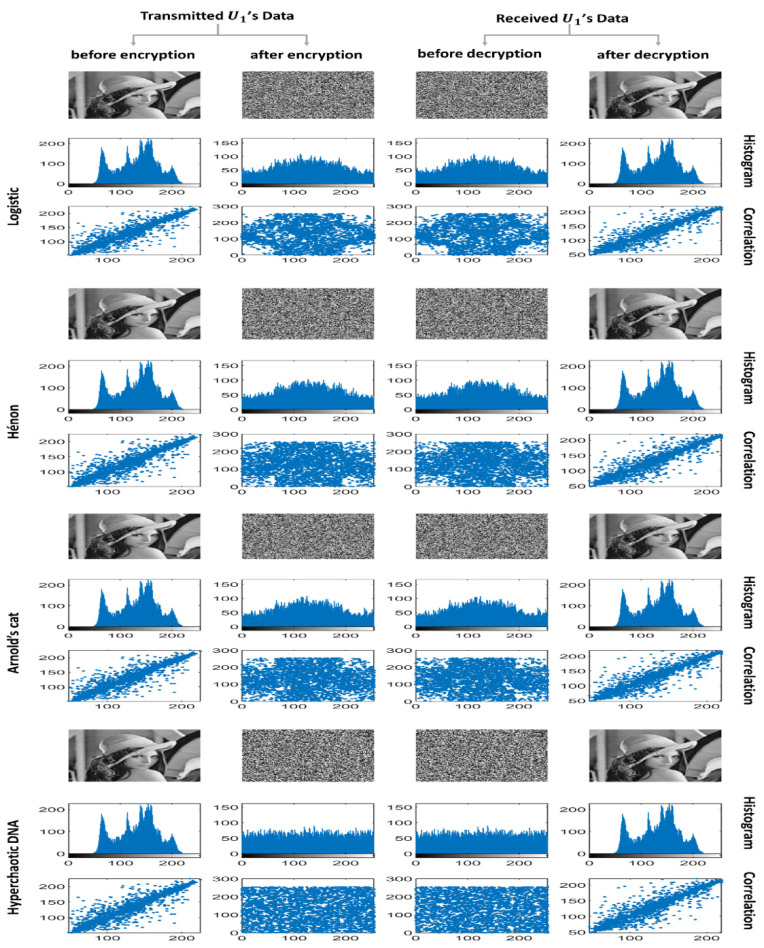
Histogram and correlation analyses.

**Figure 9 entropy-25-00140-f009:**
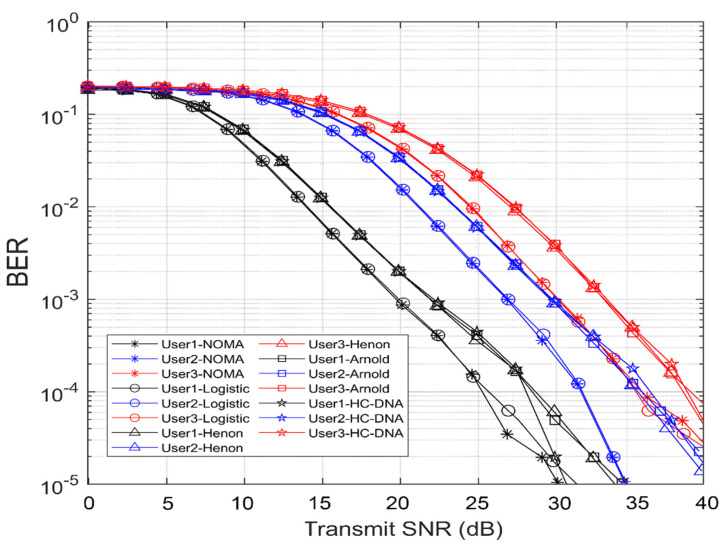
BER performance for the four C-DL-NOMA schemes with three users.

**Figure 10 entropy-25-00140-f010:**
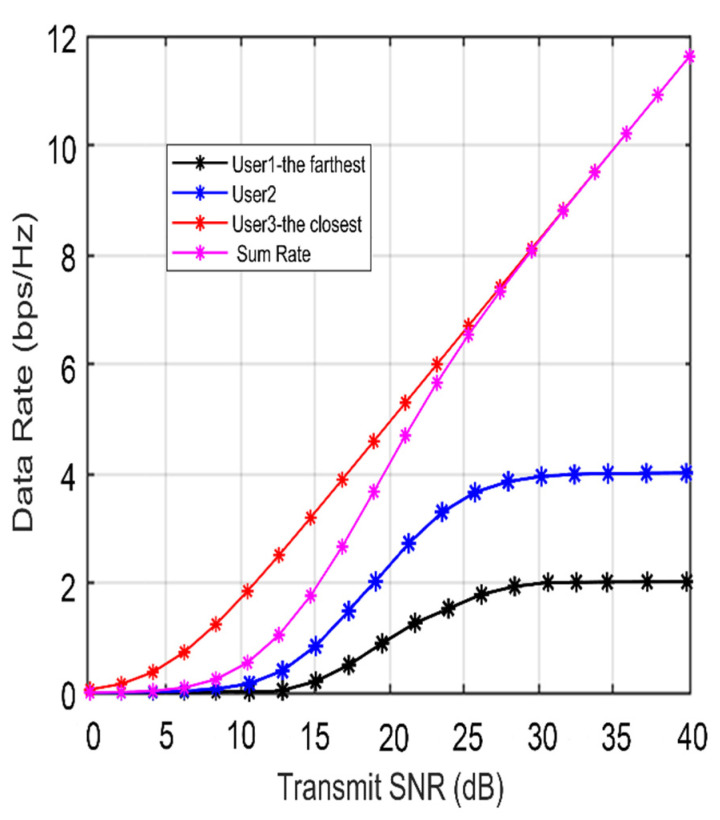
Achievable data rates and sum rates for the three users in C-DL-NOMA.

**Table 1 entropy-25-00140-t001:** Simulation parameters.

Monte Carlo Simulation	MATLAB Programming
User Number	$\mathrm{Three}\;\mathrm{Users}\;(U_{1}$ $,\;U_{2}$ $,\;U_{3})$
Data Type/Size	Image Message/(128 × 128)
Bandwidth	1 MHZ
Transmit SNR	0 to 40 dB
Thermal Noise Density	Log_10_(kT) = −174 dBm/Hz
Path Loss Exponent	4
Modulation Scheme	QPSK
Channel Type	AWGN and Rayleigh-Fading Channel
Number of Antennas	Tx = 1, Rx = 1
Power Allocation Factors	$a_{1}$ = 0.62$,\;a_{2}$ = 0.3, $a_{3}$ = 0.08
Distances (m)	$d_{1}$ = 186$,\;d_{2}$ = 90$,\;d_{3}$ = 24
Cryptographic Algorithm	Chaotic Map-Based Cryptography

**Table 2 entropy-25-00140-t002:** Information entropy of encrypted images.

	Logistic	Hénon	Arnold’s Cat	Hyperchaotic DNA
$U_{1}$ **’** **s Data**	7.9974	7.9963	7.9977	7.9964
$U_{2}$ **’** **s Data**	7.9967	7.9975	7.9969	7.9979
$U_{3}$ **’** **s Data**	7.9982	7.9984	7.9987	7.9985

**Table 3 entropy-25-00140-t003:** BER performance for the four C-DL-NOMA schemes with three users.

	User 1	User 2	User 3
	SNR (dB)	SNR (dB)	SNR (dB)
NOMA	19.6	26.8	30
Logistic	19.8	26.9	30.1
Hénon	22	29.5	33.3
Arnold’s Cat	22	29.5	33.3
Hyperchaotic DNA	22.18	29.6	33.6
Asif Mahmood [14]	26	30.5	-
BER	$10^{-3}$	$10^{-3}$	$10^{-3}$

## Data Availability

Not applicable.

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
