# Peer review of "Physical-Layer Security in Power-Domain NOMA Based on Different Chaotic Maps"

_entropy, 2023, doi:10.3390/e25010140_

Round 1

Reviewer 1 Report

The paper is written and organized well. However, there are some improvements which are needed before the paper may be accepted for publication.  

1. Writing and style improvements:

i. Insert a space in line 59 “security [4][5].Several”.

ii. Some acronyms are undefined e.g., RC4, SISO.

iii. Table 1, some words are small while others are capital. Please be consistent.

iv. Line 95-101 contributions shall be in the form of bullet points.

v. Indent shall be removed after equation (1) and other places.

2. System Model Fig1 and Fig2 is not visually clear and not generic. It is advised to make the system model more generic instead of for three users.

3. Line 130: specific type of fading shall be included in the text.

4. In (13) some in-between steps of the two equalities shall be included.

5. In (15), does * indicate convolution or multiplication. Please clarify.

6. Pseudo code shall be written starting with what are inputs and outputs and then followed by the process for description in line 203-209, 231-236, 255-260, 293-309, and 318- 336.

7. Validation of closed-form expressions of performance metrics is missing in the paper. It is suggested to include figures of analytical and simulation overlapping results.

8. More literature on Physical layer security on NOMA such as https://doi.org/10.3390/s21124180 may be added if relevant.

Author Response

We’d like to thank the reviewer for his time and precious comments. Overall, we think that his recommendations increase the quality of presenting our proposals. In the following, we present the changes that have been done to the manuscript.

Reviewer 2 Report

Specifics

The number inside the parenthesis indicates the row that the comment is about.

(53) There should be references to support the claim that traditional encryption techniques are not effective in devices with such constraints.

(128) Do not use the letter “i” in the denominator, since it can be confused with the letter “i” from the nominator and the “i” on the left side of the equation.

Equation (6) seems to be the achievable rate for decoding the signal x_{i}, and not x_{j}, as stated in the text.

(163) The meaning of the sentence “transmission of SNR” is not clear. Would it be “SNR during transmission”?

There are two Figure 3 in the manuscript.

(212) Be careful with the sentences here. First, you say the system cannot be chaotic, unless the values for a and b are 1,4 and 0,3, respectively. Then you say that the system can be chaotic for other values. The problem is with the first sentence. Those values provide the classic Hénon map, which is chaotic. However, the system can be chaotic for other values, as you say in your second sentence. Hence, the two sentences are saying opposing things.

Equations (16) and (17) appear as floating elements in the text. Equations are not floating elements, but part of the text, even when they appear in dedicated rows. Here, I suggest doing as it was done for equation (15), which was preceded by text.

(215) This sentence appears random. It should be explained what "Fixed point" and "the slopes" are.

(220) This is not Figure 1, this is Figure 4. Be careful with captions.

(231) This sentence does not explain very well what is being done. How pixels can be extracted only by measuring the height and width of an image?

(235) When stating “input image”, it is not clear if it is the original image or the scrambled image resulting from the operation with the Hénon map. Please be specific.

At this point, I would say that the description of the encryption and decryption processes are presented in a very confusing way. I would suggest using a pseudo-algorithm structure to present in a logical order, using variables to show that the output of a procedure is used as an input to another. Another way to illustrate that could be using a block diagram.

(241) It does not change randomly. One of the features of this map is that it appears random, but this is different from being actually random.

In section 3 it is not clear whether the base station knows the channel response or if it has to estimate it prior in the simulation. I think it was considered perfectly known at the base station, which is reasonable. However, it is important information and should be explicit in the text.

The Histogram analysis can be improved. From the description in the text, the closer the histogram is to a uniform distribution, the more secure the encryption mechanism. Although visually is very obvious that the hyperchaotic map is closer to a uniform distribution than the others, the authors might still perform a fitting to the uniform distribution. This way, the other maps can also be compared, and not only be said that they perform worse than the hyperchaotic map.

(393) How is P(Ti) calculated? Does Ti really denote the encrypted image, or it denotes a sequence of 8 bits whose probability depends on how many pixels denoted by this specific sequence there are in the image?

(402) First, the authors say that the closer user (User 3) has the worst performance. Then, in row (404), they say User 3 performs best. Also, the reason for performing better is not clear. If user 3 performs no SIC, should it not perform worse, since it considers a bigger part of the signal as noise? Also, from Figure 8, User 3 is shown to have the worst performance. Hence, the BER analysis in this first moment is all wrong. The user with the best performance is User 1, which obtained the lowest BER.

Also, the figure that should be Figure 8, is shown as Figure 4.

(407) What does explain the behavior of the Henon and Arnold mechanisms? Do they not use the logistic map to generate the key values? Why are they different from the logistic map-only performance? What causes this?

(410) What does Table 3 shows? What does the SNR in this table represent? Why are they different for different techniques? How are they obtained? The BER shown is referring to what encryption technique? Or does it show the SNR necessary to obtain a 1e-3 BER? This table should be better explained and discussed in the text.

(417) When referring to equations, the number should be inside parenthesis.

The outage probability analysis is not related to the security of the system. At least not the way it was developed in this article. Thus, if the paper is about physical layer security, as the title indicates, there was no need to perform such analysis. It is possible, however, to include the physical layer security aspect in the analysis, which I would recommend doing if it is worth the effort. Nonetheless, it does not allow any conclusions about the security of the system.

Overall comments

I have big concerns regarding the contributions of the work. It seems to me that what was done was to apply encryption and decryption techniques based on chaotic maps in a power domain NOMA system. Those encryption mechanisms could be employed in any wireless communication system. They could even be employed in the application layer. The physical layer security should take advantage of some features from the physical layer, such as channel fading. That is not the case, however. If so, it is not clear from the text what are the advantages of using the NOMA for the signals encrypted with the techniques presented. Here, the NOMA system and the chaotic encryption system work independently.

The fact is that a NOMA system influences the achievable rate analysis, but that is it. And the way it was developed has nothing to do with security. My suggestion is that the paper focuses on encryption mechanisms as the main topic, and for evaluation, a NOMA system will be employed to transmit the encrypted information.

Another concern is regarding the chaotic encryption and decryption techniques. Are those techniques developed by the authors? Or are they found in the literature? It is not clear from the text where they came from. Another point is that it seems that, except for the hyperchaotic map, the others techniques end up using the logistic map to encrypt and decrypt the message. The maps are being used as an additional scrambling method.

Author Response

(The authors gave the same response as above.)

Reviewer 3 Report

There are many editing errors, e.g.:

- there is no space after the dot ending the sentence in line 59, p.2

- two references to papers are listed one by one, but they should be merged into a single set o brackets in the same line

- some of the equations (eg. eq. 7) and figures (e.g. fig. 1) are too big and use margin space

- the "ith" in line 132 is not a correct English word

- the numbering of figures is repeating the same numbers twice - there two figures 1

The DOIs in references need to be double-checked, as e.g. in [12] the DOI points to the editorial of the volume, not to the article with title matching the title of the paper cited.

So many abbreviations are used on pages 1 and 2 that it is hard to read the text. Maybe some of them could be removed if they are not used later on?

The division of three sentences into three paragraphs in lines 95-101 is also hard to read. This should instead be merged into a single passage.

The system model presented on page 3 on figure 1 suggests that the solution is limited to a fixed number of 3 users, but the equations presented below show that this is not the case.

The equation on page 4 are in large part repeating the analysis which has been done in [12], this should be clearly stated in the paper and a clear description of what is novel and what has been repeated from previous work should be given.

The simulation results prove that the proposed method is working correctly and the method is showing interesting results, however, due to the many errors listed above it is hard to follow the description of the work.

Author Response

(The authors gave the same response as above.)

Round 2

Reviewer 1 Report

Authors have addressed my comments well.

Author Response

We’d like to thank the reviewer for his time and precious comments. Overall, we think that his recommendations increase the quality of presenting our proposals.

Reviewer 2 Report

1. Reference [6] should have the conference name on it.

2. The concern about the letter "i" is regarding another equation, that unfortunately is not numbered, and is now on line 133. Yes, the “i” is the user number. But in this equation, which denotes the normalized power factor, it is also used as iterator in the sum in the denominator of the fraction, which is confusing. The suggestion is to use another letter here, so the reader do not confuse the "i" that indicates the user number and the "i" that is the iterator in the sum. I know the subscript is the user number, but the reader might substitute the value "i" on the left side of the equation and also substitute the same number in the "i" in the denominator. This recommendation is just for the sake of formality.

3. Now it seems to be the achievable rate for decoding the signal x_{m} and not x_{j}. In fact, the variable “j” does not even appear in equation (6) anymore. Should the sum in the denominator start in “j+1” instead of “m+1”?

4. Perfect.

5. Perfect

6. The issue was fixed, but the phrasing is still confusing. I would suggest something in the lines of “The classical Hénon map uses the values a=1.4 and b =0.3, which results in a chaotic behavior.”

7. Issue fixed.

8. Issue fixed.

9. Issue fixed.

10. Perfect

11. Perfect

12. Perfect.

13. Ok.

14. Perfect.

15. Perfect

16. Perfect

17. Ok

18. Ok.

19. It is now clear what this table represents. I would suggest reviewing the English in this added text, though. There are expressions such as “to can” that is not correct.

20. Ok

21. Yes, and that is my point. The encryption schemes do not affect the outage probability because it considers the input and the output of the channel. More precisely, it only considers the already encrypted information, which is the input of the channel, and the information before decryption, which is the output of the channel.

And the curves for the different schemes are not shown, just the curves for different users. So it is not really showing that different encryption schemes impact the outage probability.

And even though the outage probability is important in the NOMA systems, that is not the goal of the presented paper. Or at least, in the first moment, is not what it appears to be. What it appears to be is the security of these systems using chaotic encryption. If one would use outage probability to evaluate the security of the system, the signals to be considered in the analysis should be the ones before encryption and after decryption. This way, the outage probability would truly show the impact of the encryption schemes.

My suggestion here is to either exclude this part of the paper, or make it clear to the reader that the way the outage probability is derived here, it does not include the encryption mechanisms in the mathematical analysis. I say this because including the encryption schemes in the mathematical analysis is difficult. However, the way it is shown now, it is impossible to conclude that the encryption schemes does not affect the outage probability. And to say that when the BER results shows differences, makes it more ambiguous. At this point, this is my major concern regarding this paper.

In summary, the problem is:

The way the outage probability analysis was conducted, does not allow the evaluation of different encryption schemes. It only evaluates the channel impact in the transmitted signal, which is already encrypted. Therefore, it is impossible to claim that different encryption schemes does not impact the outage probability between the information before encryption and after decryption. It is possible to see that this is not true, since different SNR values are required to achieve the same level of BER.

Author Response

(The authors gave the same response as above.)

Reviewer 3 Report

All comments from the previous review have been appropriately addressed.

Author Response

(The authors gave the same response as above.)

Round 3

Reviewer 2 Report

The authors have dealt with all my concerns.